# Differential systemic inflammatory responses after TAVI: The role of self versus balloon expandable devices

**Haitham Abu Khadija**[○], **Gera Gandelman**[○], **Omar Ayyad, Mustafa Jaber, Lion Poles, Michael Jonas, Offir Paz, Firas Abu Sbaih, Gal Sella, Sara Shimoni, Jacob George**\*, **Alex Blatt**[○]\*

Heart Center, Kaplan Medical Center, Rehovot, Affiliated with the Hebrew University, Jerusalem, Israel

[○] These authors contributed equally to this work.

\* alexbl31@clalit.org.il (AB); jgeorge@bezeqint.net. (JG)

**Data Availability Statement:** All relevant data are within the paper.

**Funding:** The author(s) received no specific funding for this work.

## Abstract

### Objective

Transcatheter aortic valve implantation (TAVI) provokes early injury response, represented in part by dynamic changes in the inflammatory markers. The association of self-expanding valves (SEVs) and balloon-expandable valves (BEVs) with the consequent inflammatory response remains uncertain.

### Materials and methods

Patients with severe symptomatic aortic stenosis who underwent transfemoral TAVI: SEVs or BEVs, from January 2010 to December 2019 were enrolled. Whole white blood cells (WBC) and subpopulation dynamics as well the neutrophil to lymphocyte ratio (NLR) were evaluated.

### Results

Three-hundred seventy consecutive patients (mean age 81.75 ± 6.8 years, 199 women's) were enrolled. In the entire population, significant kinetic changes in the WBC response (p <0.0001) between admission and first 24 hours post procedure, with a significant increase in total WBC (7.46 ± 2.26 to 10.08 ± 3.55) and absolute neutrophil count (4.97 ± 2.06 to 8.19 ± 3.43), NL ratio (3.72 ± 2.8 to 9.76 ± 7.29), and a meaningful decrease in absolute lymphocytes count (1.67 ± 1.1 to 1.1 ± 0.76). When compared between the types of valves, SEVs were associated with a more pronounced inflammatory response than BEVs, with total WBC (10.44 ± 3.86 vs. 9.45 ± 3.19) neutrophils (8.56 ± 3.75 vs. 7.55 ± 3.06) with p 0.016 and 0.012 respectively.

### Conclusion

This is the first description of a differential inflammatory response between the two leading delivery systems. SEV appears to trigger a more robust inflammatory response as

**Competing interests:** The authors have declared that no competing interests exist.

compared to BEV. Clinical studies are warranted to assess the long term effect of our findings.

## Introduction

Previous studies have reported that developing a systemic inflammatory response syndrome (SIRS) after transcatheter valve implantation (TAVI) is associated with poor prognosis [1–7]. In recent years, much effort has gone into improving existing prognostic models for risk stratification of patients undergoing TAVI. The neutrophil to lymphocyte ratio (NLR) is a well described prognostic marker in cardiovascular medicine [8–11]. There is no clinical data comparing the impact of self-expandable valves (SEV) and balloon-expandable valves (BEV), the two leading implantation systems on NLR. The objective of our study was to investigate the differences among these two systems with respect to the value and kinetic behavior of white blood cells (WBC) and their subsets after TAVI and compare the short-term outcomes of these procedures.

As inflammation triggers a series of biological processes that could potentially promote the long term viability of the valves, it is important to study differences in the pro-inflammatory behavior of different valves and associate those with long term durability and outcome.

## Materials and methods

### Study participants and TAVI procedure

We retrospectively included patients with severe symptomatic aortic stenosis who underwent trans-femoral TAVI in our center between January 2010 and December 2019. This study approved by the local Helsinki committee, Kaplan Medical Center, confirmation number 0091-20-KMC. The data used was completely anonymized, de-identified, and aggregated before access and analysis.

We excluded patients who did not have repeated blood tests and those with periprocedural death (up to 72 hours after TAVI). The type and size of the valves were at the discretion of the local heart team after they assembled, and a decision was made according to the patient's anatomical and clinical characteristics. Patients treated with BEV were implanted with either Sapien, Sapien XT, or S3 (Edwards Lifesciences, Irvine, California) valves. Patients treated with SEV were implanted with either Corevalve, Evolute-Pro, or Evolute-R (Medtronic, Inc., Minneapolis, Minnesota) valves. The small number of subjects treated with other valves was also excluded from the analysis. Trans-femoral vascular access and closure was performed using the percutaneous approach with the safety wire technique and the Prostar XL (Abbott Vascular, Redwood City, California) vascular closure device. The procedure duration was calculated as "skin to skin", i.e. time 0 was the opening of the arterial blood pressure from the accessory support access and the final time was represented by the closure of this accessory support access. Since then, we also used local anesthesia with conscious sedation as a first-line approach. All patients received unfractionated heparin to maintain a minimum active clotting time >250 seconds after the insertion of the femoral sheet. Protamine (1 mg for each 100 U of heparin, maximal dose 50 mg) was administrated at the time of vascular closure if needed. The use of prophylactic antibiotics during the procedure or hospital stay was routinely avoided. Aspirin was recommended before TAVI. Dual-antiplatelet treatment with aspirin 100 mg and clopidogrel 75 mg was started the day before the procedure and followed thereafter for six months, except for patients requiring chronic oral anticoagulation.

### Blood samples, inflammatory markers and definition criteria for events

Baseline characteristics, procedural data, and clinical outcomes were collected. Blood samples were obtained using a 21G sterile syringe without stasis. Laboratory analyses were performed before the procedure, during the patient's postprocedural intensive care unit stay on a daily basis, and at the physician's discretion in the cardiology ward, and were retrospectively collected. Neutrophil-Lymphocyte Ratio (NLR) was calculated using this formula: [NLR = absolute neutrophils count/absolute lymphocytes count]. The definition of the patients' baseline characteristic risk factors included the corresponding diagnosis in the medical records together with the chronic use of anti-hypertensive, glycemic control and lipid lowering drugs, respectively. Coronary artery disease (CAD), atrial fibrillation, peripheral arterial disease and strokes, were based on the medical records and confirmed by complementary clinical tools, such as angiograms in CAD. All the outcome definitions were strictly determined according the Valve Academic Research Consortium 2 Criteria (VARC-2). Standard follow-up included 30-day and six-month visits after hospital discharge. This follow-up was performed on site.

### Statistical analysis

Data are presented as means ± standard deviations. Continuous variables between the various study groups were tested for normality by a Shapiro- Wilk test and when an abnormal distribution was found, a Mann- Whitney test was performed. When the distribution was normal, a t-test was used. Pearson's chi-square test was performed for categorical variables when appropriate. Main effect estimates are presented with their 95% confidence interval.

We divided the patients into two groups according to the TAVI expandable system. One group was patients receiving SEV and the other were patients receiving BEV. Repeated measures analysis of variance was used to determine any significant differences between variability over time between SEV and BEV. The Kaplan-Meier method was used for cumulative survival analysis after six months. To compare the survival between patients that received SEV TAVI and patient that received BEV TAVI, a log-rank test was used when appropriate. A value of $P < 0.05$ was defined as statistically significant. Analyses were performed using IBM Statistical Package for the Social Sciences (SPSS) version 23.0 (IBM Corp., Armonk, NY).

## Results

### Patients' and procedural characteristics

During the ten years study period, 370 consecutive patients were enrolled. The study flowchart is shown in Fig 1. Only 22 patients were excluded. The analyzed population included 348 patients (57.2% female, mean age 81.7 ± 6.8 years) with severe, symptomatic aortic stenosis (mean transaortic pressure gradient 45.0 ± 13.9 mmHg), and high or prohibitive operative risk (STS score of 8.01 ± 1.5). Baseline and procedural characteristics of the study population according to types of expandable systems are summarized in Table 1.

There were no significant differences between the two group types at baseline and no differences in procedural characteristics.

### Inflammatory markers dynamics

Baseline total WBC, absolute cell counts of neutrophils, monocytes and lymphocytes, and neutrophil to lymphocytes ratio (NL Ratio) and their dynamic changes after TAVI for the total study population are summarized in Table 2. In the entire population, we noticed that there were significant kinetic changes in the WBC response (*p* value <0.0001) between

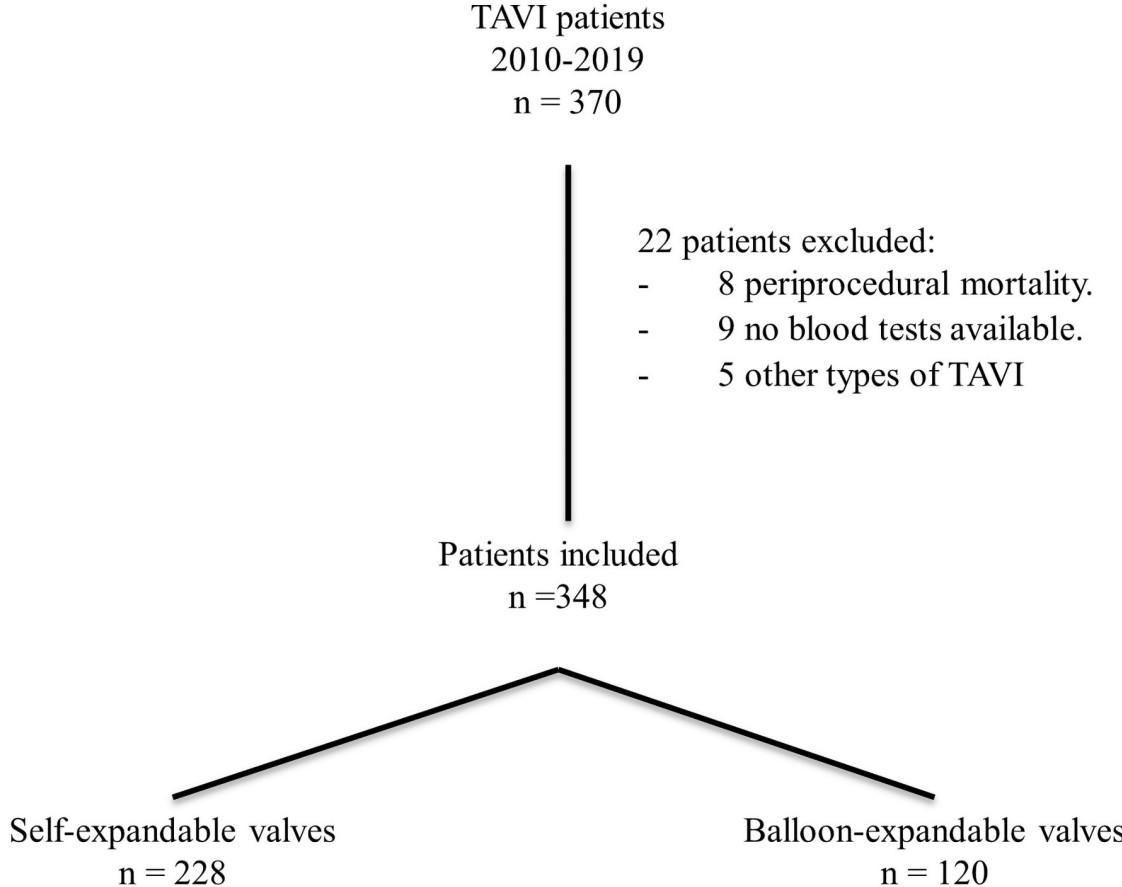

**Fig 1. Study flowchart of patients from 2010 to 2019.** A total of 370 patients were treated with TAVI. After excluding 22 patients, a total of 348 patients were finally included in the analysis.

admission, 24 hours and 72 hours post procedure, with significant increases in WBC, neutrophils absolute, monocytes absolute, NL ratio, and significant decreases in absolute lymphocyte count.

Fig 2 shows the dynamic changes of white blood cells and its differential subsets with time, in the first 72 hours post-procedure. There is a significant increase in inflammatory markers including total WBC, neutrophils, monocytes and NL ratio.

When comparing between the two valve systems as shown in Fig 3, SEVs were associated with a more pronounced inflammatory response than BEVs, with total WBC 10.44 ± 3.86 vs. 9.45 ± 3.19 neutrophils 8.56 ± 3.75 vs. 7.55 ± 3.06 with p value 0.016 and 0.012 respectively. The Kaplan-Meier survival curve at the ten-year follow up is shown in Fig 4. During the ten-year study period, we found no significant differences between the two types of expandable systems (p value 0.949).

The variables that may be related to an increase in inflammatory markers after TAVI are presented in Table 3. In the variable analysis, the duration of the procedure and the amount of contrast used were related to a heightened inflammatory response with a p value < 0.01. Septum thickness at baseline (13.55 ± 2.04) was associated with a higher risk for inflammatory response post-procedure with a significant p value.

**Table 1. Patients' and procedural characteristics.**

| | Overall (N = 348) | Balloon expandable valve (N = 120) | Self-expandable valve (N = 228) | p value |
|---|---|---|---|---|
| Clinical Characteristic | | | | |
| Age, years | 81.75 ± 6.85 | 81.92 ± 6.28 | 81.66 ± 7.14 | 0.744 |
| Women (%) | 199 (57.2%) | 67 (55.8%) | 132 (57.9%) | 0.798 |
| Body mass index, kg/m2 | 28.20 ± 5.14 | 28.44 ± 5.17 | 28.07 ± 5.13 | 0.522 |
| Hypertension (%) | 321 (92.2%) | 113 (94.2%) | 208 (91.2%) | 0.445 |
| Diabetes (%) | 144 (41.5%) | 51 (42.5%) | 93 (41.0%) | 0.872 |
| Dyslipidemia (%) | 262 (75.3%) | 91 (75.8%) | 171 (75.0%) | 0.968 |
| Smoker (%) | 33 (9.5%) | 11 (9.2%) | 22 (9.7%) | 1 |
| Atrial Fibrillation (%) | 100 (29.1%) | 36 (30.0%) | 64 (28.6%) | 0.878 |
| Coronary artery disease (%) | 138 (39.9%) | 40 (33.6%) | 98 (43.2%) | 0.108 |
| Peripheral artery disease (%) | 44 (12.7%) | 15 (12.6%) | 29 (12.8%) | 1 |
| Previous myocardial infarction(%) | 33 (9.5%) | 6 (5.0%) | 27 (11.9%) | 0.059 |
| Previous stroke (%) | 28 (8.0%) | 5 (4.2%) | 23 (10.1%) | 0.085 |
| Previous pacemaker (%) | 43 (12.4%) | 14 (11.8%) | 29 (12.7%) | 0.933 |
| CABG (%) | 16 (7.3%) | 7 (9.6%) | 9 (6.1%) | 0.511 |
| STS Score | 8.01 ± 1.5 | 8.04 ± 1.45 | 7.95 ± 1.62 | 0.617 |
| Medications | | | | |
| Aspirin (%) | 221(63.5%) | 74(61.6%) | 147(64.4%) | 0.720 |
| P2Y12 inhibitors (%) | 93(26.7%) | 23(19.1%) | 70(30.7%) | 0.281 |
| DAPT therapy (%) | 55 (15.8%) | 18(15.0%) | 37 (16.2%) | 0.890 |
| Statin (%) | 254 (72.9%) | 84 (70.0%) | 170 (74.5%) | 0.621 |
| Glucophage (%) | 114 (32.7%) | 40 (33.3%) | 74 (32.4%) | 0.848 |
| Insulin (%) | 88 (25.2%) | 25 (20.8%) | 63 (27.6%) | 0.504 |
| Steroids (%) | 33 (9.4%) | 9 (7.5%) | 24 (10.9%) | 0.162 |
| Anticoagulant (%) | 84 (24.1%) | 30 (25.0%) | 54 (23.6%) | 0.696 |
| ACE-I / ARBs (%) | 293 (84.1%) | 95 (79.1%) | 198 (86.8%) | 0.732 |
| Beta Blockers (%) | 198 (56.8%) | 74 (61.6%) | 124 (54.3%) | 0.663 |
| Echocardiography | | | | |
| LVEF (%) | 52.8 ± 9.8 | 52.5 ± 10.1 | 53.5 ± 9.3 | 0.408 |
| Septum thickness (mm) | 13.55 ± 2.04 | 13.45 ± 2.03 | 13.6 ± 2.06 | 0.501 |
| Aortic valve area (cm2) | 0.69 ± 0.16 | 0.71 ± 0.15 | 0.67 ± 0.15 | 0.393 |
| Aortic valve gradient (mm Hg) | 45.0 ± 13.9 | 45.6 ± 15.1 | 44.0 ± 11.6 | 0.704 |
| Blood count | | | | |
| WBC (K/uL) | 7.51 ± 2.35 | 7.50 ± 2.15 | 7.51 ± 2.46 | 0.951 |
| Absolute Neutrophils (K/uL) | 5.03 ± 2.16 | 5.02 ± 1.86 | 5.04 ± 2.30 | 0.91 |
| Absolute Lymphocytes (K/uL) | 1.66 ± 1.06 | 1.61 ± 0.69 | 1.68 ± 1.22 | 0.575 |
| NLR | 3.78 ± 2.82 | 3.82 ± 2.62 | 3.76 ± 2.93 | 0.844 |
| Procedure details | | | | |
| Contrast volume (ml) | 134.79 ± 59.84 | 129.67 ± 57.54 | 137.48 ± 60.96 | 0.25 |
| Time (minutes) | 93.19 ± 32.83 | 92.86 ± 31.84 | 93.36 ± 33.40 | 0.895 |

Values are mean ± SD or n (%).

ACE-I = Angiotensin Converting Enzyme Inhibitor; ARBs = Angiotensin Receptor Blockers; CABG = coronary artery bypass; STS = Society of Thoracic Surgeons; LVEF = left ventricle ejection fraction; WBC = white blood cells; NLR = Neutrophils to Lymphocytes Ratio.

**Table 2. Dynamic changes of WBC and their subpopulation after TAVI.**

| | Admission | 24h | 72h | 6 Month | p1 | p2 | p3 |
|---|---|---|---|---|---|---|---|
| WBC (K/uL) | 7.46 ± 2.26 | 10.08 ± 3.55 | 9 ± 2.91 | 7.47 ± 2.38 | <0.0001 | <0.0001 | 1.000 |
| Absolute Neutrophils (K/uL) | 4.97 ± 2.06 | 8.19 ± 3.43 | 6.73 ± 2.79 | 4.89 ± 2.04 | <0.0001 | <0.0001 | 1.000 |
| Absolute Lymphocytes (K/uL) | 1.67 ± 1.1 | 1.1 ± 0.76 | 1.29 ± 0.59 | 1.76 ± 1.09 | <0.0001 | <0.0001 | 1.000 |
| Absolute Monocytes (K/uL) | 0.55 ± 0.24 | 0.65 ± 0.44 | 0.74 ± 0.37 | 0.33 ± 0.92 | 0.004 | <0.0001 | 0.715 |
| NLR | 3.72 ± 2.8 | 9.76 ± 7.29 | 6.52 ± 4.66 | 3.36 ± 2.23 | <0.0001 | <0.0001 | 1.000 |

Values are mean ± SD.

p1 = Comparison of pre-procedural values with those at 24h.

p2 = Comparison of pre-procedural values with those at 72h.

p3 = Comparison of pre-procedural values with those at 6 months.

WBC = white blood cells; NLR = Neutrophils to Lymphocytes Ratio.

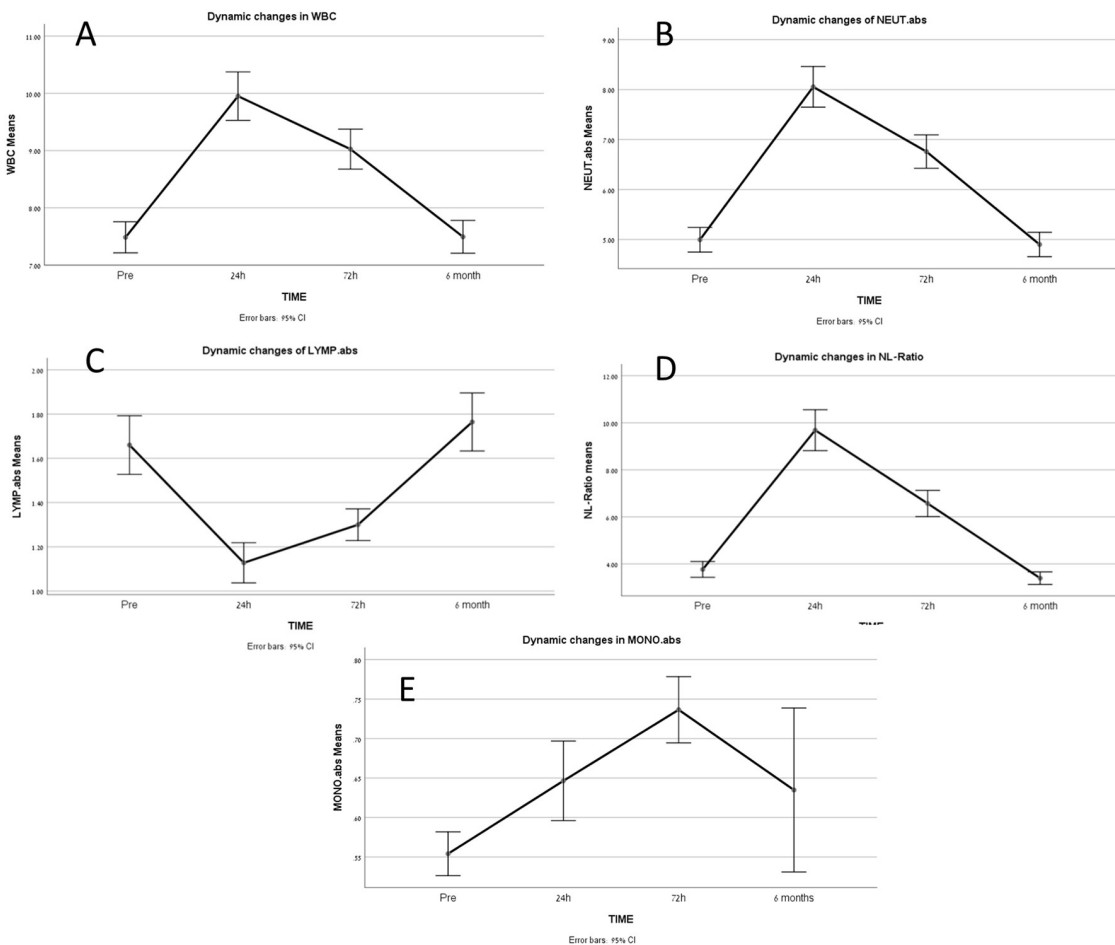

**Fig 2. Dynamic changes of leukocytes and its components after TAVI, (A) dynamic changes in total WBC with time, (B) dynamic changes in Neutrophils with time, (C) dynamic changes of lymphocytes with time, (D) dynamic changes of NL ratio with time, (E) dynamic changes of monocytes with time.** LYMP. abs = Absolute Lymphocytes (K/uL); NEUT. abs = Absolute Neutrophils (K/uL); MONO. abs = Absolute monocytes (K/uL); WBC = White Blood Cells (K/uL); NLR = Neutrophils to Lymphocytes Ratio.

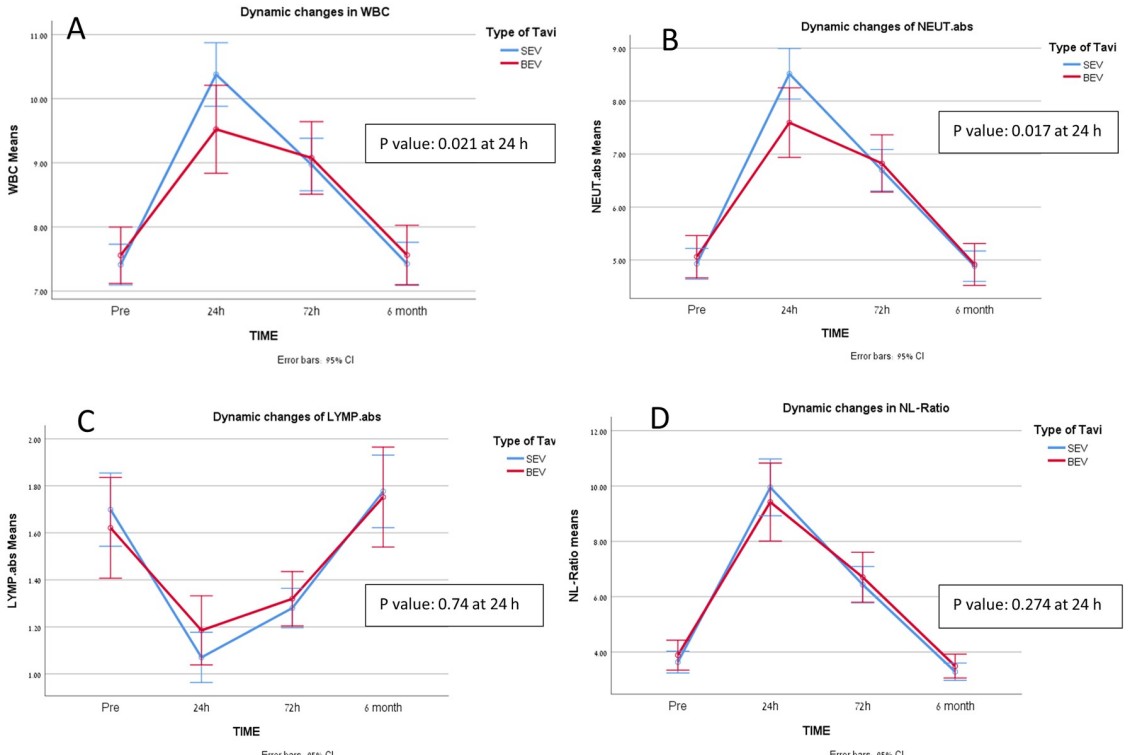

**Fig 3. Difference in two types of expandable system for leukocytes and its components after TAVI, (A) difference in total WBC with time, (B) difference in Neutrophils with time, (C) difference in lymphocytes with time, (D) difference in NL ratio with time.** SEV = Self Expandable Valve; BEV = Balloon Expandable Valve; LYMP. abs = Absolute Lymphocytes (K/uL); NEUT. abs = Absolute Neutrophils (K/uL); WBC = White Blood Cells (K/uL); NLR = Neutrophils to Lymphocytes Ratio.

### Inflammatory response according the valve delivery system

We found a positive significant association between the types of the expandable system and inflammatory markers, with SEV implantation predicting an elevated inflammatory response as also demonstrated in Fig 3.

### Clinical outcomes

The thirty-day clinical outcomes according to the Valve Academic Research Consortium-2 criteria are summarized in Table 4. An increase in WBC post-TAVI was associated with higher rates of major bleeding, arrhythmia and mortality at 30 days (Table 5).

### Discussion

We observed an acute increase in leukocytes and their subpopulations, and consequently, increased NLR after TAVI. This response was also found to be associated with a worse 30 days outcome. SEV is associated with more leukocytosis, more lymphopenia and a higher NLR.

SIRS has been described in different cardiovascular scenarios such as acute coronary syndromes, heart failure and interventional procedures, including TAVI. In this last setting, a comprehensive understanding of the underlying biological responses to the prosthesis is essential to corroborate with potential future complications. These triggers comprise of three main pathogenic pathways leading to SIRS' clinical manifestations: activation of leucocytes and other inflammatory components, endothelial injury, and myocardial/pericardial injury.

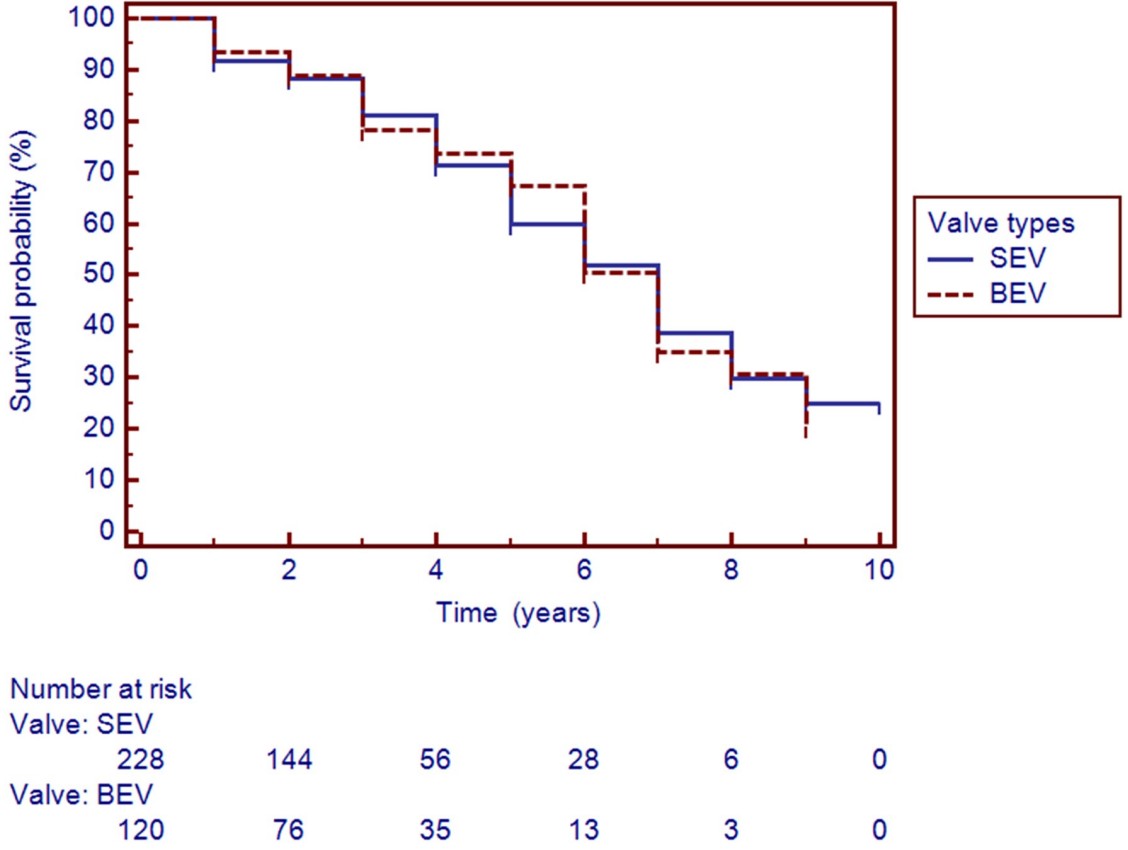

**Fig 4. Kaplan Meir curve based on all cause mortality for patients underwent TAVI, showing no difference in survival between the two TAVI types during ten years follow up.** SEV = Self Expandable Valve; BEV = Balloon Expandable Valve.

In this work, we studied different inflammatory markers that may potentially affect clinical outcome. We found that there is a more than an 80% increase in total leukocyte count, neutrophil count and a decrease in lymphocyte count, in the first 72 hours post TAVI in our study population. Post-TAVI, 47% of the patients had total WBC and an absolute neutrophils counts of more than 11 K/uL and 8, 11 K/uL, respectively. As recently described by Baratchi et al. [12] monocytes have a pivotal role in the chronic process and contribute to aortic stenosis development. We found a significant rise in the monocytes count in the acute phase, but we did not measure their activated function. This limits the possible interpretations.

The neutrophil to lymphocyte ratio (NLR) has been proposed as an accurate prognostic marker of a systemic inflammatory response in previous studies [8–11]. In our study, 57% of the study population had an NLR of more than nine which is considered moderate to high stress and a robust inflammatory marker.

Neutrophils mediate some of the inflammatory responses to acute injury by numerous biochemical mechanisms, resulting in further tissue damage. These include the release of arachidonic acid metabolites and platelet-aggravating factors, cytotoxic oxygen–derived free radicals [13], myeloperoxidase [14], elastase [15], various hydrolytic enzymes such as acid phosphatases [16], etc. The relative lymphopenia observed post TAVI can be explained by the endogenous cortisol segregation response to the procedure's stress [17].

Our results are in agreement with those obtained by Condado, et al. [18], who reported that an elevated NLR is associated with the occurrence of composite adverse outcomes at 30 days

**Table 3. Variable analysis of factors that may influence inflammatory markers at 24 hours post TAVI †.**

| | | WBC | | Abs. Neutrophils | | Abs. Lymphocytes | | NLR | |
|---|---|---|---|---|---|---|---|---|---|
| | | *(mean ± SD)* | *p* value | *(mean ± SD)* | *p* value | *(mean ± SD)* | *p* value | *(mean ± SD)* | *p* value |
| Total | | 10.101 ± 3.671 | | 8.216 ± 3.559 | | 1.095 ± 0.729 | | 9.855 ± 7.534 | |
| Baseline characteristics | | | | | | | | | |
| Gender | Women | 10.151 ± 3.798 | 0.795 | 8.282 ± 3.713 | 0.568 | 1.144 ± 0.863 | 0.273 | 9.811 ± 7.873 | 0.838 |
| | Men | 10.033 ± 3.504 | | 8.125 ± 3.35 | | 1.029 ± 0.491 | | 9.913 ± 7.081 | |
| Age*, years | | 0.024 | 0.652 | 0.012 | 0.823 | -0.014 | 0.8 | 0.035 | 0.518 |
| Body mass index*, kg/m2 | | 0.101 | 0.062 | 0.077 | 0.152 | 0.127 | 0.018 | -0.004 | 0.937 |
| Hypertension | no | 10.266 ± 4.814 | 0.791 | 8.578 ± 4.657 | 0.576 | 0.946 ± 0.434 | 0.258 | 10.682 ± 6.417 | 0.593 |
| | yes | 10.086 ± 3.567 | | 8.184 ± 3.458 | | 1.107 ± 0.747 | | 9.785 ± 7.625 | |
| Diabetes | no | 10.111 ± 3.705 | 0.877 | 8.323 ± 3.65 | 0.459 | 1.055 ± 0.681 | 0.315 | 10.237 ± 7.833 | 0.474 |
| | yes | 10.095 ± 3.645 | | 8.073 ± 3.443 | | 1.148 ± 0.792 | | 9.354 ± 7.098 | |
| Dyslipidemia | no | 10.253 ± 3.479 | 0.77 | 8.446 ± 3.272 | 0.449 | 1.021 ± 0.452 | 0.684 | 9.796 ± 5.786 | 0.875 |
| | yes | 10.05 ± 3.736 | | 8.139 ± 3.65 | | 1.118 ± 0.798 | | 9.874 ± 8.034 | |
| Smoker | no | 10.077 ± 3.706 | 0.909 | 8.202 ± 3.605 | 0.974 | 1.09 ± 0.741 | 0.915 | 9.879 ± 7.628 | 0.698 |
| | yes | 10.332 ± 3.42 | | 8.381 ± 3.175 | | 1.134 ± 0.623 | | 9.75 ± 6.761 | |
| Atrial Fibrillation | no | 10.172 ± 3.682 | 0.643 | 8.294 ± 3.545 | 0.565 | 1.081 ± 0.659 | 0.949 | 9.823 ± 7.281 | 0.915 |
| | yes | 9.952 ± 3.684 | | 8.034 ± 3.627 | | 1.14 ± 0.887 | | 9.837 ± 8.163 | |
| Coronary artery disease | no | 10.312 ± 3.968 | 0.215 | 8.363 ± 3.868 | 0.38 | 1.142 ± 0.856 | 0.309 | 10.112 ± 8.58 | 0.343 |
| | yes | 9.768 ± 3.145 | | 7.983 ± 3.011 | | 1.018 ± 0.474 | | 9.496 ± 5.637 | |
| Peripheral artery disease | no | 10.125 ± 3.689 | 0.911 | 8.232 ± 3.586 | 0.956 | 1.1 ± 0.756 | 0.59 | 9.895 ± 7.643 | 0.842 |
| | yes | 10.041 ± 3.63 | | 8.192 ± 3.457 | | 1.063 ± 0.532 | | 9.709 ± 6.974 | |
| Previous myocardial infarction | no | 10.173 ± 3.726 | 0.446 | 8.275 ± 3.629 | 0.507 | 1.104 ± 0.755 | 0.46 | 9.962 ± 7.748 | 0.467 |
| | yes | 9.647 ± 2.857 | | 7.834 ± 2.647 | | 1.024 ± 0.402 | | 8.987 ± 5.151 | |
| Previous stroke | no | 10.098 ± 3.668 | 0.945 | 8.21 ± 3.555 | 0.918 | 1.095 ± 0.745 | 0.785 | 9.906 ± 7.654 | 0.649 |
| | yes | 10.132 ± 3.764 | | 8.275 ± 3.661 | | 1.085 ± 0.508 | | 9.27 ± 6.08 | |
| Previous pacemaker | no | 10.112 ± 3.746 | 0.992 | 8.209 ± 3.619 | 0.84 | 1.109 ± 0.75 | 0.288 | 9.632 ± 7.253 | 0.163 |
| | yes | 10.09 ± 3.138 | | 8.319 ± 3.151 | | 0.997 ± 0.562 | | 11.472 ± 9.286 | |
| CABG | no | 9.913 ± 3.533 | 0.305 | 8.025 ± 3.435 | 0.359 | 1.119 ± 0.703 | 0.387 | 9.31 ± 7.272 | 0.966 |
| | yes | 8.977 ± 3.062 | | 7.215 ± 2.79 | | 0.963 ± 0.479 | | 9.23 ± 5.84 | |
| STS Score* | | 0.134 | 0.423 | 0.312 | 0.52 | 0.114 | 0.81 | 0.135 | 0.518 |
| Baselines Echo | | | | | | | | | |
| LVEF (%) * | | 0.1 | 0.065 | 0.008 | 0.139 | 0.106 | 0.048 | 0.015 | 0.799 |
| AVA (cm2) * | | -0.023 | 0.689 | -0.03 | 0.602 | 0.074 | 0.201 | -0.025 | 0.656 |
| AV Gradient mean (mm Hg) * | | 0.036 | 0.698 | 0 | 0.999 | 0.108 | 0.236 | -0.014 | 0.881 |
| LVEDD (mm) * | | -0.024 | 0.664 | -0.014 | 0.798 | -0.051 | 0.346 | 0.047 | 0.389 |
| Septum thickness (mm) * | | 0.118 | 0.029 | 0.108 | 0.045 | 0.001 | 0.985 | 0.08 | 0.14 |
| MR-Pre | other | 10.020 ± 3.443 | 0.598 | 8.064 ± 3.292 | 0.33 | 1.116 ± 0.522 | 0.537 | 9.019 ± 6.098 | 0.019 |
| | moderate to severe | 10.233 ± 3.991 | | 8.445 ± 3.907 | | 1.066 ± 0.946 | | 11.057 ± 9.072 | |
| Procedure related | | | | | | | | | |
| TAVI types | SEV | 10.442 ± 3.859 | 0.016 | 8.564 ± 3.751 | 0.012 | 1.068 ± 0.514 | 0.353 | 10.096 ± 7.463 | 0.411 |
| | BEV | 9.451 ± 3.197 | | 7.552 ± 3.068 | | 1.145 ± 1.02 | | 9.396 ± 7.676 | |
| Contrast volume (ml)* | | 0.192 | <0.001 | 0.209 | <0.001 | -0.068 | 0.205 | 0.22 | <0.001 |
| Time (minutes)* | | 0.204 | <0.001 | 0.243 | <0.001 | -0.172 | <0.001 | 0.33 | <0.001 |
| Post dilatation | no | 10.129 ± 3.684 | 0.785 | 8.214 ± 3.573 | 0.926 | 1.115 ± 0.769 | 0.156 | 9.702 ± 7.365 | 0.28 |
| | yes | 9.95 ± 3.681 | | 8.208 ± 3.567 | | 0.996 ± 0.515 | | 10.643 ± 8.373 | |
| Post TAVI Echo | | | | | | | | | |
| AV Gradient (mm Hg) * | | 0.058 | 0.305 | 0.073 | 0.195 | -0.004 | 0.947 | 0.037 | 0.512 |

*(Continued)*

**Table 3.** (Continued)

| | WBC | | Abs. Neutrophils | | Abs. Lymphocytes | | NLR | |
|---|---|---|---|---|---|---|---|---|
| | (mean ± SD) | p value | (mean ± SD) | p value | (mean ± SD) | p value | (mean ± SD) | p value |
| AI Post (moderate)* | 0.012 | 0.865 | 0.017 | 0.812 | -0.132 | 0.06 | -0.017 | 0.81 |

† Data are presents as means ± standard deviations. Continues variables between the various study groups were tested for normality by Shapiro- Wilk test and when abnormal distribution was found Mann- Whitney test was performed, when the distribution was normal t—test was used.

* Correlation between two Continues variables was tested by Pearson Correlation.

TAVI = transcatheter valve implantation; CABG = coronary artery bypass; STS = Society of Thoracic Surgeons; LVEF = left ventricle ejection fraction; AVA = aortic valve area; AV = aortic valve; LVEDD = left ventricular end diastolic diameter; MR = Mitral Regurgitation; SEV = self-expandable valve; BEV = balloon expandable valve; AI = aortic insufficiency; WBC = white blood cells; Abs = absolute; NLR = Neutrophils to Lymphocytes Ratio.

in patients undergoing balloon expandable TAVI. We extended this observation to SEV as well as BEV systems comparatively.

Our findings suggesting a link between post TAVI peri-nflammatory response and poorer 30 days outcome is similar to a previous study by Schwietz T, et al. [3]. They describe that the occurrence of SIRS in the first 48 hours post-procedure is associated with impaired prognosis following TAVI, but they included both transfemoral and transapical access routes while our findings related only to the femoral approach. The transapical approach is associated with significantly more organ trauma and consequent inflammatory response that could have clouded the results. Despite the statistically significant differences between the two types of valves at 24 hours post-procedure, this is not reflected in the survival rate. This is similar to other clinical situations where laboratory significant changes are seen without a clinical translation. Similarly, thrombocytopenia after TAVI is a universal and virtually inevitable phenomenon in more than 90% of the patients [19]. However, a clinical impact is only found when there is an

**Table 4. Association of inflammatory markers at 24 hours and thirty days outcome in patient underwent TAVI †.**

| | | WBC | | Abs. Neutrophils | | Abs. Lymphocytes | | NLR | |
|---|---|---|---|---|---|---|---|---|---|
| | | (mean ± SD) | p value | (mean ± SD) | p value | (mean ± SD) | p value | (mean ± SD) | p value |
| Total | | 10.1 ± 3.671 | | 8.216 ± 3.559 | | 1.095 ± 0.729 | | 9.855 ± 7.534 | |
| 30 days mortality | NO | 9.997 ± 3.544 | 0.008 | 8.11 ± 3.427 | 0.005 | 1.098 ± 0.734 | 0.609 | 9.69 ± 7.39 | 0.014 |
| | YES | 14.503 ± 6.082 | | 12.686±5.974 | | 0.925 ± 0.433 | | 16.857 ± 10.423 | |
| Bleeding | NO | 9.979 ± 3.541 | 0.093 | 8.098 ± 3.424 | 0.072 | 1.082 ± 0.634 | 0.469 | 9.587 ± 7.111 | 0.005 |
| | YES | 11.485 ± 4.778 | | 9.552 ± 4.716 | | 1.231 ± 1.434 | | 12.917 ± 11.016 | |
| Major vascular complication | NO | 9.982 ± 3.537 | 0.098 | 8.115 ± 3.452 | 0.18 | 1.107 ± 0.749 | 0.85 | 9.545 ± 7.099 | 0.067 |
| | YES | 11.46 ± 4.605 | | 9.369 ± 4.367 | | 1.007 ± 0.539 | | 12.872 ± 10.566 | |
| Stroke | NO | 10.067 ± 3.526 | 0.705 | 8.171 ± 3.405 | 0.74 | 1.1 ± 0.734 | 0.098 | 9.697 ± 7.058 | 0.608 |
| | YES | 10.541 ± 6.07 | | 8.985 ± 6.071 | | 0.891 ± 0.496 | | 13.483 ± 14.152 | |
| MI | NO | 10.081 ± 3.665 | 0.419 | 8.192 ± 3.545 | 0.448 | 1.094 ± 0.732 | 0.511 | 9.78 ± 7.274 | 0.844 |
| | YES | 11.442 ± 4.241 | | 9.778 ± 4.584 | | 1.152 ± 0.556 | | 14.99 ± 19.13 | |
| AKI | NO | 10.046 ± 3.624 | 0.527 | 8.165 ± 3.509 | 0.573 | 1.086 ± 0.729 | 0.365 | 9.728 ± 6.903 | 0.525 |
| | YES | 11.044 ± 4.402 | | 9.086 ± 4.342 | | 1.242 ± 0.727 | | 12.053 ± 14.844 | |
| Arrhythmia | NO | 10.02 ± 3.762 | 0.375 | 8.07 ± 3.599 | 0.151 | 1.15 ± 0.815 | 0.023 | 9.205 ± 6.696 | 0.017 |
| | YES | 10.291 ± 3.453 | | 8.561 ± 3.452 | | 0.963 ± 0.44 | | 11.401 ± 9.077 | |

† Data are presents as means ± standard deviations. Continues variables between the various study groups were tested for normality by Shapiro- Wilk test and when abnormal distribution was found Mann- Whitney test was performed, when the distribution was normal t—test was used

AKI = acute kidney injury; Abs = absolute; NLR = Neutrophils to Lymphocytes Ratio; WBC = white blood cells.

**Table 5. Thirty-days outcomes of patients after TAVI, SEV vs. BEV †.**

| Variable | Total | SEV | BEV | p Value |
|---|---|---|---|---|
| | (n = 348) | (n = 228) | (n = 120) | |
| 30 days mortality | 8 (2.3%) | 7 (3.1%) | 1 (0.8%) | 0.271 |
| Bleeding | 28 (8.0%) | 19 (8.3%) | 9 (7.5%) | 0.786 |
| Major vascular complication | 33 (9.5%) | 22 (9.6%) | 11 (9.2%) | 0.884 |
| Stroke | 16 (4.6%) | 11 (4.8%) | 5 (4.2%) | 0.774 |
| Myocardial infarction | 5 (1.4%) | 4 (1.8%) | 1 (0.8%) | 0.663 |
| Acute Kidney Injury | 19 (5.5%) | 14 (6.1%) | 5 (4.2%) | 0.441 |
| Arrhythmia | 103 (29.6%) | 71 (31.3%) | 32 (26.7%) | 0.385 |

† Categorical and Nominal variables were reported by prevalence and percentages and were analyzed by pearson's chi-square ($\chi^2$) test and Fisher's Exact Test.

extreme thrombocyte count drop. In our present study, we did not discriminate according to the inflammatory response intensity and the clinical outcomes, which probably can shed light this finding.

The procedure risk score assessment is a dynamic process, it will change according the clinical evolution. We propose to add our findings, as they are inexpensive to do, to the well-established and known pre-TAVI procedure scores, such as CAPRI. We have demonstrated that a simple, routine test WBC count and its differential is a potentially valuable screening tool for risk stratification of patients who undergo TAVI. The association between unfavorable 30 day outcomes of deaths, bleeding and vascular access complications, according to the VARC-2 definition, highlights that the more complex procedures provoke major inflammatory responses. Therefore, less favorable outcomes or inherently, a major inflammatory response predisposes the patient to unfavorable clinical events related to the existing milieu. The addition of NLR to the TAVI risk score improved the predictability of MACE after TAVI. This cost-effective routine test can possibly guide or even justify upcoming anti-inflammatory therapies in patients with severe aortic stenosis post TAVI if a cause and effect relationship is firmly established between inflammation and poorer outcome.

To the best of our knowledge, we are the first to describe a difference between the two leading TAVI valve systems that impose different mechanical stresses on the vessel wall and have different cusp origins. SEV is associated with a higher pro-inflammatory response as compared with the BEV system. The underlying explanation can be ascribed to the differences in the delivery systems. BEV's frame is comprised of cobalt chromium, and the leaflet component is based on bovine tissue whereas the SEV is made of a nitinol scaffold and the cusps-leaflet component is porcine pericardial tissue. Adverse responses to xenogeneic pericardium-based valves might be prevented by tissue decellularization, ideally removing all cells and preserving the original extracellular matrix. A comparative assessment of acellular pericardia from different species is still lacking, as opposed to the commonly implanted glutaraldehyde (GA) preserved native tissues. Submitting native tissues to a decellularization process causes different effects on these tissues in terms of their histological, immunohistochemical, biochemical, and ultrastructural properties, as well as their denaturation, biomechanical, and cytocompatibility profiles [20]. Additionally, the decellularization process can degrade matrix components, which result in loss of extracellular matrix integrity [21]. The resultant tissue deterioration can lead to degenerative structural graft failure [21]. A major concern with porcine-derived materials is the presence of residual cells, DNA, and the alpha-Gal epitope after the decellularization process. These issues justify the ongoing efforts to achieve a reliable and safe tissue-engineered heart valve (TEHV).

Endothelial injury and foreign body placement lead to the activation of platelets at the implant site with recruitment of circulating leukocytes. The ultimate biocompatibility of a device will be influenced by both inflammation and coagulation. Studies have found that the adsorption of fibrinogen and platelet activation to nitinol surfaces is dependent on the surface chemical composition and topography. Specifically, titanium content enhances the adsorption of fibrinogen [22, 23]. With respect to metal ion delivery, studies have shown that cobalt-chromium and stainless-steel alloys tend to release less nickel than nitinol in Hank's solution when the surface finishing processes are similar [24]. These observations may explain the differences that we observed with regard to the differential inflammatory response to the two valvular systems.

One key to a successful TAVI procedure is the safe anchoring of the prosthesis and the alignment to the patient's anatomy. The TAVI valve prosthesis anchoring mechanism relies entirely on oversizing that provokes different degrees of pressure and consequently subjacent tissue injury. Therefore, a radial force exists between the surrounding anatomy and the prosthesis. The influence of the delivery system and mechanical anchoring forces is the most remarkable difference between these valves. The BEV tissue injury surface is less than that of SEV due to BEV's "ring" design and the "sleeve" shape which has more contact surface than SEV. Furthermore, the nitinol shape memory behavior of the SEV continues to expand after implantation. Thus, the apparently increased tissue stress imposed by the SEV system could account for the heightened inflammatory response as compared to the BEV system.

The use of iodinated contrast agents has also been proposed as another possible etiologic factor for the enhanced systemic inflammatory response. Their chemical properties, immunoallergic reactions, and genetic predispositions, are some of the possible explanations to understand this relationship. We presume that the association between the inflammation, procedure duration and amount of contrast agents are all surrogates of the complexity of the procedure more than a mechanistic cause. Some confounders, such as the peri-procedure use of anti-inflammatory drugs, including steroids in iodine contrast allergy mitigation was minimal and with a similar rate in the two groups, thus there is no bias.

We recognize certain limitations of this study. Being a single-center retrospective observational study is in itself a partial limitation. However, methodologically, the fact that only one team performed the procedure allowed for treatment uniformity to be maintained. Additionally, our sample size, although large, was not robust and therefore did not allow for sufficient power to detect overall mortality event rates.

In conclusion, to the best of our knowledge, this is the first report that found major differences in the early post-procedural inflammatory responses to TAVI, with SEV being associated with a heightened response as compared with BEV. These observations should be further pursued in larger studies as it may have a potentially detrimental effect on the long-term durability of these biological valves.

## Author Contributions

**Conceptualization:** Lion Poles.

**Data curation:** Gera Gandelman, Gal Sella.

**Formal analysis:** Omar Ayyad.

**Methodology:** Mustafa Jaber, Gal Sella.

**Resources:** Michael Jonas, Firas Abu Sbaih.

**Software:** Firas Abu Sbaih.

**Supervision:** Offir Paz.

**Validation:** Sara Shimoni.

**Visualization:** Sara Shimoni, Jacob George.

**Writing – original draft:** Haitham Abu Khadija, Offir Paz, Jacob George, Alex Blatt.

**Writing – review & editing:** Haitham Abu Khadija.

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
