## [Decision Letter · Decision Letter 0]

28 Apr 2021

PONE-D-21-04417

Differential systemic inflammatory responses after TAVI: The role of self versus balloon expandable devices

PLOS ONE

Dear Dr. Blatt,

Thank you for submitting your manuscript to PLOS ONE. After careful consideration, we feel that it has merit but does not fully meet PLOS ONE’s publication criteria as it currently stands. Therefore, we invite you to submit a revised version of the manuscript that addresses the points raised during the review process.

Your manuscript has been carefully evaluated by four external reviewers and the editorial team as a Regular Article. In its present form the manuscript is not acceptable for publication.

Although all reviewers and editors found that this manuscript has merit and addresses a significant clinical problem, the reviewers raise some important questions. These comments include direct comparison in short-term SIRS between SEV and BEV (reviewer#1), other inflammatory markers (reviewers #2 and #3), medications (reviewer#2), and more detailed discussion based on the authors’ findings and metal differences between SEV and BEV (reviewers #1, #3 and #4).

The authors should address all comments that the reviewers raised. In addition to the reviewers' comments, there are several issues that the authors should address.

1)Survival rates are similar between patients with SEV and BEV, suggesting the minimum impact of TAVI-induced inflammatory response on the survival. Relevance of increased WBC and neutrophils only 24 hours post procedure should be provided. At a minimum, discuss the relevance more in detail.

2)Describe the selection process for BEV and SEV.

3)Provide information on medications including statins, other lipid lowering drugs, and anti-diabetic drugs.

4)Consider potential confounders, including medications.

5)Provide more detailed methods, including determination of study size and definition of diagnosis (hypertension, diabetes, dyslipidemia, smoker, atrial fibrillation, coronary artery disease, etc) and all outcomes (bleeding, major vascular complication, stroke, myocardial infarction, AKI, and arrhythmia).

6)Describe the time point when inflammatory markers (blood samples) were obtained post TAVI; 24 hr 72 hr or 6 months, in Table 3.

7)Explain the data on table 3 and 4 more in detail with the method of statistical analysis in the main text and table legend.

8)Describe the method of statistical analysis in Figure 3.

9)Unites are missing in a lot of data. Provide.

10)Number at risk should be provided in Figure 4.

11)Provide more detailed figure legends. Also, add figure 4 legend.

We look forward to receiving your revised manuscript.

Kind regards,

Michinari Nakamura, MD

Academic Editor

PLOS ONE

Journal Requirements:

Please provide additional details regarding participant consent. In the Methods section, please ensure that you have specified (1) whether consent was informed and (2) what type you obtained (for instance, written or verbal). If your study included minors, state whether you obtained consent from parents or guardians. If the need for consent was waived by the ethics committee, please include this information.

Please include your tables as part of your main manuscript and remove the individual files. Please note that supplementary tables (should remain/ be uploaded) as separate "supporting information" files

Reviewers' comments:

Reviewer's Responses to Questions

**Comments to the Author**

1. Is the manuscript technically sound, and do the data support the conclusions?

Reviewer #1: No

Reviewer #2: Partly

Reviewer #3: Yes

Reviewer #4: Partly

2. Has the statistical analysis been performed appropriately and rigorously? 

Reviewer #1: No

Reviewer #2: Yes

Reviewer #3: Yes

Reviewer #4: Yes

3. Have the authors made all data underlying the findings in their manuscript fully available?

Reviewer #1: No

Reviewer #2: Yes

Reviewer #3: Yes

Reviewer #4: Yes

4. Is the manuscript presented in an intelligible fashion and written in standard English?

Reviewer #1: Yes

Reviewer #2: Yes

Reviewer #3: Yes

Reviewer #4: Yes

5. Review Comments to the Author

Reviewer #1: In their manuscript titled “Differential systemic inflammatory response after TAVI: The role of self versus balloon expandable devices”, authors tried to compare SIRS between SEV and BEV. The conclusion revealed (1) SEV trigger more SIRS than BEV groups and (2) the short term effect may be associated with long term valvular durability. However, the whole tables did not show any comparison between SEV and BEV, except table 1 (demography). The remaining table just manifested total patients result. Reader may more interest in comparison of WBC, absolute neutrophils, absolute, lymphocytes, postop complication, and short-term result between SEV and BEV groups. Although authors mentioned the difference between two groups in Figure 3, there is no comparative result to enhance the effect of SIRS between two groups, such as postop complication, short-term result. These comparative results should be listed in tables so that matched their title “SEV vs BEV”. Second, short-term result was shown on table 4 and longer result was on Figure 4 (KM curve). The big problem is table 4 is calculated from total number but figure 4 statistics is from separated group. So, I am not sure the authors can conclude “the short-term effect may be associated with long term result”. Furthermore, there isn’t any postop valve evaluation result in this article. This conclusion didn’t make sense.

As authors’ mention, any implant will induce SIRS in patients. So, if authors just show total patients’ data in WBC, neutrophil, lymphocyte, N/L ratio change after TAVI without separated groups’ comparison, it is very usual and predictable result. Many aortic stent graft paper also discuss about effect of metal material and SIRS because stent graft material is very similar to TAVI’s valve support structure, especially in metal material. So, this article should classify the total patients and focus on cobalt chromium (BEV) and nitinol (SEV) comparison. Try to discuss “different” metal material to cause “different” prognosis by “different” SIRS results and statistics.

Generally speaking, this title didn’t match their result and this result (total patients) is also not special. Study of different metal material induce SIRS is frequently published in the patients who receiving aortic stent graft procedure. Please classify to two groups by BEV and SEV groups more detail. Analyze individual pre-, post- inflammation index, perioperative parameter, postoperative complication, short-term follow-up, and valvular durability so that authors can conclude “The role of self versus balloon expandable devices”

Reviewer #2: 1. The study presents the results of original research.

Yes

2. Results reported have not been published elsewhere.

To my knowledge after a brief research the results have von been published yet.

3. Experiments, statistics, and other analyses are performed to a high technical standard and are described in sufficient detail.

Yes. The data derive from a fairly large patient cohort. Performed statistical analyses seem appropriate and are described detailed

4. Conclusions are presented in an appropriate fashion and are supported by the data.

Even though the drawn Conclusions are presented well, further analyses are needed to support them. For instance no data on use of antibiotic treatments are provided. Even though especially the NLR has been shown to be a prognostic marker in cardiovascular diseases other markers of inflammation such as C - reactive protein or procalcitonin should have been taken into consideration since the authors state themselves, that SIRS is triggered among others by “activation of leucocytes and other inflammatory components”. In order to sustainable support the message more data analyses are needed.

In the discussion the passage on leaflet components spares any support by the data and remains hypothetical. The authors should discuss the fact that procedure duration impacted inflammatory response. Furthermore, the authors do not at all discuss their findings on survival.

5. The article is presented in an intelligible fashion and is written in standard English.

The manuscript is well written.

6. The research meets all applicable standards for the ethics of experimentation and research integrity.

Since the results are derived from retrospective collected data, so from my point of view no ethic problems apply.

7. The article adheres to appropriate reporting guidelines and community standards for data availability.

Yes

Reviewer #3: Khadija et al. investigated inflammatory changes that occur after transcatheter aortic valve implantation (TAVI) in 370 patients. The authors investigated the white blood cell counts before and after the procedure (24 hours), as well as the neutrophil and lymphocyte count and their ratio. The main finding is that inflammation is increased in all patients but significantly more in of self-expanding valves compared to and balloon-expandable valves. The authors discuss several potential reasons for this but do not prevent data potentially explaining this difference.

Overall this is a focused, concise and well written paper. The data are well presented.

One of the issues is that the authors should discuss that this is not a randomized trial. For this reason there remains the caveat that there is some bias in the selection of the patients.

Do the authors have monocyte data as well? There was a recent paper in Circulation (Baratchi et al. 2000) showing that monocytes are activated before TAVI and this activation is reduced after TAVI. This finding would be of interest to be discussed in the context of the authors’ current findings.

The authors should also discuss potential clinical consequences of their findings.

Reviewer #4: In this manuscript, Khadija and colleagues retrospectively investigated the kinetic of white blood cell differential counts at different time points after TAVI to compare the pro-inflammatory and prognostic impact of self-expandable and balloon expandable valves in 348 patients undergoing transcatheter aortic valve implantation (TAVI) for severe aortic stenosis. Routine laboratory records of total leukocyte, neutrophil and lymphocyte counts, and the neutrophil-lymphocyte were analyzed prior to TAVI, post-TAVI, 24 hrs, 72 hrs, and 6 months post-TAVI.

There was a significant postprocedural increase in WBS, neutrophil counts and NLR, accompanied by significant decrease in total lymphocyte counts at 24 hours and at 72 hours, followed by normalization of parameters at 6 months after TAVI. Whereas self-expandable valves were associated with more pronounced neutrophilia, there were no significant differences in 10-year patients’ survival between the both types of expandable systems. Although the dynamic changes of WBS and neutrophils after TAVI marked by a significant increase of neutrophil counts and lymphocytic drops (and the increased NL ratio, accordingly) 24 hours after TAVI are per se not particularly surprising and have been already described previously (as already cited in the current manuscript), the actual differences between the both leading TAVI delivery systems are interesting. The authors spend a great deal of the discussion dealing with the possible pathophysiological explanations for differential inflammatory response between SEVs and BEVs. This is a very learned discussion. Overall, this study is well performed and presented. I have following comments for authors' consideration.

1. I do not agree with or do not understand the authors’ discussion statement: “The addition of NLR to the TAVI risk score improved the predictability of MACE after TAVI.” What ‘TAVI risk score’ are the authors referring to? Do the authors present such data or do they refer to any published data (please include a reference if so)? Do the authors postulate that their results outline the feasibility of leukocyte counts or NLR as a novel additive tool for improved post-procedural patient assessment? Can authors prove e.g. improvement of the AUC of the ROC curve after addition of NLR to known scores (such as STS)? The authors should clarify these issues, add new analyses supporting their statements, or tone down the above sentence.

2. Why was pre-TAVI septal hypertrophy associated with a higher risk of postprocedural neutrophilia? Possible explanation? Did patients with thicker interventricular septum have possibly higher WBS/neutrophil counts at baseline – before TAVI (being indicative for pre-existing low-grade inflammation, likely associated with IVS hypertrophy/remodeling) when compared with non-hypertrophic pts? Significant basal septal hypertrophy bears the risk of maldeployment of the TAVI valve. So the presence of LVH (or septal hypertrophy) may influence the choice of TAVI delivery system. Did the patients with IV hypertrophy receive SEVs more frequently? Might the presence of LV/septal hypertrophy have biased the results while predisposing for more pronounced neutrophil mobilization (and being possibly independent on the valve type)? Please clarify/discuss.

3. Please explain/shortly discuss the association between the VARC-2-defined 30-day outcomes and WBC elevation? Why was leukocytosis associated with increasing bleeding rates? Did the bleeding events occurred within the first 24 hours after TAVI being possibly partially causative for WBS increase (inflammation is knowingly associated with bleeding events as related to stress, hematoma organization etc.). Was there any association between delivery system (SEVs vs. BEVs) and 30-day complications according to VARC-2 criteria?

6. PLOS authors have the option to publish the peer review history of their article (what does this mean?). If published, this will include your full peer review and any attached files.

Reviewer #1: No

Reviewer #2: No

Reviewer #3: No

Reviewer #4: No

---

## [Author Response · Author response to Decision Letter 0]

18 Jul 2021

Emily Chenette, Editor-in-Chief

Sara Fuentes Pérez, Senior Editor 

PLoS ONE, Cardiovascular and Public Health

Dear Profs. Chenette and Fuentes Pérez,

Thank you for reconsidering our revised manuscript. We have made significant efforts to meet the reviewers' valuable comments.

Attached is a point-by-point letter addressing the comments and the changes made accordingly.

We are submitting (uploading) the revised new manuscript version and deleting the previous file. The new sentences have been placed as described in this letter below. The sentences in bold type are the updated changes.

We hope you will find the current version and the changes made, worthy of publication in the PLoS ONE Open Journal. 

Detailed answers to the editor and referees comments:

1) Survival rates are similar between patients with SEV and BEV, suggesting the minimum impact of TAVI-induced inflammatory response on the survival. Relevance of increased WBC and neutrophils only 24 hours post procedure should be provided. At a minimum, discuss the relevance more in detail.

Despite the statistically significant differences between the two types of valves at 24 hours post-procedure, this is not reflected in the survival rate. This is similar to other clinical situations where laboratory significant changes are seen without a clinical translation. Similarly, thrombocytopenia after TAVI is a universal and virtually inevitable phenomenon in more than 90% of the patients. However, a clinical impact is only found when there is an extreme thrombocyte count drop. In our present study, we did not discriminate according to the inflammatory response intensity and the clinical outcomes , which probably can shed light this finding.

 (These sentences were added to the main manuscript, Discussion section)

2) Describe the selection process for BEV and SEV.

The type and size of the valves were at the discretion of the local heart team after they assembled, and a decision was made according to the patient’s anatomical and clinical characteristics.

(This sentence was added to the main manuscript, Methods section)

3) Provide information on medications including statins, other lipid lowering drugs, and anti-diabetic drugs.

See new Table 1. We added the drug history for each patient which includes antiplatelet, anticoagulant, lipid-lowering agents, steroids, and medications for diabetes and hypertension.

4) Consider potential confounders, including medications.

Some confounders, such as the peri-procedure use of anti-inflammatory drugs, including steroids in iodine contrast allergy mitigation was minimal and with a similar rate in the two groups, thus there is no bias.

(This sentence was added to the main manuscript, Discussion section)

5) Provide more detailed methods, including determination of study size and definition of diagnosis (hypertension, diabetes, dyslipidemia, smoker, atrial fibrillation, coronary artery disease, etc) and all outcomes (bleeding, major vascular complication, stroke, myocardial infarction, AKI, and arrhythmia).

We performed a retrospective study, including all the patients that underwent TAVI in a 10-year period. Therefore, this is not an event rate driven study.

The definition of the patients’ baseline characteristic risk factors included the corresponding diagnosis in the medical records together with the chronic use of anti-hypertensive, glycemic control and lipid lowering drugs, respectively. Coronary artery disease (CAD), atrial fibrillation, peripheral arterial disease and strokes, were based on the medical records and confirmed by complementary clinical tools, such as angiograms in CAD. All the outcome definitions were strictly determined according the Valve Academic Research Consortium 2 Criteria (VARC-2).

(This sentence was added to the main manuscript, Methods section)

6) Describe the time point when inflammatory markers (blood samples) were obtained post TAVI; 24 hr 72 hr or 6 months, in Table 3

See the corrected new Table 3. We considered inflammatory markers at 24 hours. 

7) Explain the data on table 3 and 4 more in detail with the method of statistical analysis in the main text and table legend

We added the two groups of (yes) and (no) for each variable and we did the comparison. 

Data are presented as means ± standard deviations. Continuous variables between the various study groups were tested for normality by a Shapiro- Wilk test and when an abnormal distribution was found, a Mann- Whitney test was performed. When the distribution was normal, a t-test was used.

(This sentence was added to the main manuscript, Methods section and the table legends were changed)

8) Describe the method of statistical analysis in Figure 3.

Repeated measures analysis of variance were used to determine any significant differences between variability over time between SEV and BEV.

(This sentence was added to the main manuscript, Methods section)

9) Unites are missing in a lot of data. Provide.

We inserted all missing units.

10) Number at risk should be provided in Figure 4

This parameter was added to the new Figure 4. 

11) Provide more detailed figure legends. Also, add figure 4 legend.

All Tables and Figures legends were changed

Detailed answers to all the referee comments:

Reviewer #1:

1. In Materials and Methods, “We excluded patients……….those with periprocedural death (up to 72 hours after TAVI)”, why you choose 72 hours as your exclusion criteria but not 24 hours? 48 hours? 96 hours or even one month?? Could you offer any reference???

We excluded these kind of patients due to the lack of blood samples and follow-up.

2. This paper discussed about inflammation response after TAVI. Although their TAVI protocol mentioned about Aspirin 100mg before procedure and dual-antiplatelet with Aspirin after that, some patients may use Aspirin, anti-platelet agent, or statin for their long-term underlying disease before procedure. Those medication can inhibit different inflammatory pathway level. So, please list patient medication in the demography and compare whether it is significant difference between two groups. Medication may affect postoperative SIRS result.

We added in Table 1 the patients’ chronic medication list and the use of steroids in iodine allergic patients just before the procedure. Also, we added in the Discussion section, a reference to the drugs (see point 4 above).

3. How do authors select BEV or SEV methods for the patient treatment?? Is there any criteria?

See point 2 above.

4. In table 3, it is very disorganized. Author may try to identify which factors (variables) may influence inflammatory markers after TAVI. However, they should organize these variables by different statistics methods. Some variables are continuous but some are categoric. If X and Y (inflammation markers) are continuous variables, please use linear regression and organize as one table. Some X belongs to categoric (hypertension, DM, TAVI type…….), authors may consider to classify inflammation markers (Y variables) to binary variable by certain cut point and use univariate logistic regression to make another table. In this manuscript, if authors can prove TAVI type (SEV and BEV, categoric variables) is one factor for influence of inflammation markers (have been classified to binary variables), that will be very significant finding and match this manuscript’s title. Anyway, separate to two tables by different statistics methods may be easier for reader understanding.

According to the reviewer’s comments, we clarified and consequently changed this table format.

5. In table 3 again, if contrast volume and time may be the factors to influence inflammatory markers, TAVI type should be evaluated with the two factors together by multivariate logistic regression to make sure whether TAVI type (SEV and BEV) is associated with inflammatory marker changes.

The study focused on the inflammatory response. We have done a multivariable analysis for these items, but we considered them to be not relevant to this article. 

6. In discussion, it may have to be re-written because manuscript result can’t support authors’ conclusion enough. Most of discussion sentences were written or quoted by other paper result. I will suggest author reorganize whole article after recalculating convincible tables and figures so that authors can say something to correspond their title.

As a result of all the major changes requested by the editors and four rewrites, the article was significantly changed, mainly in the discussion section. We wrote an original paper that included ideas and conclusions from the literature to support or refute our hypothesis which is findings based. 

Reviewer #2:

1. For instance no data on use of antibiotic treatments are provided

The use of prophylactic antibiotics before or after TAVI procedure is not routinely recommended (Gomes B, Periprocedural antibiotic treatment in transvascular aortic valve replacement. J Interv Cardiol. 2018 Dec;31(6):885-890). To clarify this point, we added a sentence to the Methods section of this manuscript: 

The use of prophylactic antibiotics during the procedure or hospital stay was routinely avoided.

2. ” other markers of inflammation such as C - reactive protein or procalcitonin should have been taken into consideration”. 

The CRP values measured in 62 patients was 3.2 ± 4.2 mg/L at baseline and 9.8 ± 8.1 mg/L 24 hours after the procedure (p < 0.001). These values were obtained in a minority of the patients and were not performed in all the timeline sequences (at admission, after 24, 48, 72 hours, and at 6 months). We did not do any procalcitonin assays. Therefore, we did not include this data in the manuscript. If the reviewer or editors insist, we can add this point in the study limitations paragraph.

3. In the discussion the passage on leaflet components spares any support by the data and remains hypothetical

According our findings, the inflammatory response quantified by neutrophils, lymphocytes and N/L ratio, is greater for SEV compared to BEV. The differences in the materials for the components, design and deployment system can hypothetically explain this significant finding. 

Reviewer #3:

1. Do the authors have monocyte data as well?

All the monocyte behavior are given as neutrophils and lymphocytes. As you can see in Table 2 and Figure 2, included in the new article version, the findings are amazing as the other components described in the manuscript. We have not given any explanations for this finding because monocytes are not part of the SIRS response and we did not perform an assessment of their activation status, a crucial point when analyzing their possible pathophysiological role.

As recently described by Baratchi et al., Circulation. 2020;142:1092–1105, monocytes have a pivotal role in the chronic process and contribute to aortic stenosis development. We found a significant rise in the monocytes count in the acute phase, but we did not measure their activated function. This limits the possible interpretations.

(This sentence was added to the main manuscript, Discussion section)

Reviewer #4:

1. I do not agree with or do not understand the authors’ discussion statement: “The addition of NLR to the TAVI risk score improved the predictability of MACE after TAVI.” What ‘TAVI risk score’ are the authors referring to?

We agree that including our findings in the weighting of existing TAVI scores such as the French CAPRI, American TAVR or UK scores, was not explained enough.

The procedure risk score assessment is a dynamic process, it will change according the clinical evolution. We propose to add our findings, as they are inexpensive to do, to the well-established and known pre-TAVI procedure scores, such as CAPRI.

(This sentence was added to the main manuscript, Discussion section)

2. Why was pre-TAVI septal hypertrophy associated with a higher risk of postprocedural neutrophilia? Possible explanation?

This finding is interesting and relatively surprising. One hypothetically explanation can be based on the hemodynamics and shear stress in the setting of a severe aortic stenosis. Because the scope of the present study focuses on inflammation response, we did not develop this finding. 

3. Please explain/shortly discuss the association between the VARC-2-defined 30-day outcomes and WBC elevation? Why was leukocytosis associated with increasing bleeding rates?

The association between unfavorable 30 day outcomes of deaths, bleeding and vascular access complications, according to the VARC-2 definition, highlights that the more complex procedures provoke major inflammatory responses. Therefore, less favorable outcomes or inherently, a major inflammatory response predisposes the patient to unfavorable clinical events related to the existing milieu.

(This sentence was added to the main manuscript, Discussion section)

---

## [Decision Letter · Decision Letter 1]

27 Sep 2021

PONE-D-21-04417R1Differential systemic inflammatory responses after TAVI: The role of self versus balloon expandable devicesPLOS ONE

Dear Dr. Blatt,

Thank you for submitting your manuscript to PLOS ONE. After careful consideration, we feel that it has merit but does not fully meet PLOS ONE’s publication criteria as it currently stands. Therefore, we invite you to submit a revised version of the manuscript that addresses the points raised during the review process.

The reviewers commented favorably on your manuscript, but had some worthwhile suggestions. The authors should address the remaining issues. Suggestions are all good ones and very constructive, which can be addressed by discussion as study limitations.

In the abstract, limitation regarding a lack of long-term effect on inflammation and outcomes should be added, such as “long-term effect needs to be investigated” or “clinical studies are warranted to assess the long term effect”. The conclusion in the abstract, “this short term effect may be associated with long term valvular durability” is not supported by your data and should be deleted.

The sentence, “an increase in WBC … was associated with higher rates of major …” (result section (clinical outcomes)) and the sentence, “this response was also found to be associated with a worse 30 days outcome”(Discussion (first paragraph)) should be de-emphasized since there is no difference or tendency in 30-day outcome (table 5).

Small sample number should be included in the limitation.

We look forward to receiving your revised manuscript.

Kind regards,

Michinari Nakamura, MD

Academic Editor

PLOS ONE

Journal Requirements:

Reviewers' comments:

Reviewer's Responses to Questions

**Comments to the Author**

1. If the authors have adequately addressed your comments raised in a previous round of review and you feel that this manuscript is now acceptable for publication, you may indicate that here to bypass the “Comments to the Author” section, enter your conflict of interest statement in the “Confidential to Editor” section, and submit your "Accept" recommendation.

Reviewer #1: (No Response)

Reviewer #3: All comments have been addressed

Reviewer #4: All comments have been addressed

2. Is the manuscript technically sound, and do the data support the conclusions?

Reviewer #1: No

Reviewer #3: Yes

Reviewer #4: Yes

3. Has the statistical analysis been performed appropriately and rigorously? 

Reviewer #1: No

Reviewer #3: Yes

Reviewer #4: Yes

4. Have the authors made all data underlying the findings in their manuscript fully available?

Reviewer #1: Yes

Reviewer #3: Yes

Reviewer #4: Yes

5. Is the manuscript presented in an intelligible fashion and written in standard English?

Reviewer #1: Yes

Reviewer #3: Yes

Reviewer #4: Yes

6. Review Comments to the Author

Reviewer #1: In their manuscript titled “Differential systemic inflammatory response after TAVI: The role of self versus balloon expandable devices”, authors tried to compare SIRS between SEV and BEV. In the revised version, author still didn’t respond several reviewer’s question accurately. Although the most critical group outcome have been shown (table 5), it didn’t reveal any significant different between SEV and BEV. Hence, it means 24 hr inflammation response difference (Table 3 and Figure 3) can’t affect the outcome between SEV and BEV according to Figure 4 and Table 5. This paper just can say authors found higher inflammatory response in SEV but can’t arbitrarily conclude this difference is associated postoperative outcome, either short-term or long-term. In other words, operator may not worry about postoperative care even higher inflammation response in SEV was found clinically because the outcome is similar to BEV.

Meanwhile, author used a lot of previous paper to explain higher inflammation response may get poor result in discussion. However, these previous evidences can’t correspond and echo this paper because SEV and BEV didn’t show significant difference in postoperative outcome in current data (Table 5). The better discussion for the result of this paper is about why higher inflammation response was found in SEV instead of outcome comparison. In current revised discussion, author mentioned about the material difference between SEV and BEV, including metal part and bovine valve decellularization. They should list more factors may affect inflammation response, such as fibrinogen level, D-dimer level, nitinol concentration and so on. That will be more reasonable to the topic of this paper.

The question as below:

1. In table 2 and table 3, authors listed 24hr WBC, neutrophil, lymphocyte, and NLR. However, the values are a little different (ex: WBC 10.08±3.55 in table 2 vs 10.101±3.671). Do they calculate from the same groups???

2. In discussion: “Our results are in agreement……..which probably can shed light this finding” may be deleted.

Although authors want to focus on the inflammation response, they described too many outcome-related sentence in the discussion. Please remove these sentence and match original result to avoid readers confusion.

3. In discussion, “To the best of our knowledge…….heightened inflammatory response as compared to the BEV system” is good and reasonable to paper result. If author can list the table compare with nitinol concentration, D-dimer, fibrinogen…..so on between SEV and BEV, these biochemistry data will be more convincible and can more focus on inflammation response. That will be the real first report.

4. How do authors select BEV or SEV methods for the patient treatment??? Authors didn’t answer directly. The selection bias may associate post-inflammation response. Please describe and clarity.

5. Contrast and surgical time are very significant difference after 24hr postoperatively to preoperatively. Contrast amount and prolonged surgical time may induce cytokine secretion. Please compare with contrast and surgical time between SEV and BEV. It may explain the inflammation response different between groups.

6. What about prior aortic valve replacement surgery between SEV and BEV (valve-in-surgical valve)??

7. How many valves were used (rescued) during surgery if the first deployed valve was malfunctional (valve-in-valve)??

8. What about reintervention rate between two groups (valve-in-valve)??

Reviewer #3: The authors have addressed my previous comments. Overall the manuscript has now been successfully improved towards a ultimate exclusive of platelets. The discussion has also been improved. The text overall would need a proper English nurse.

Reviewer #4: Overall, the authors have improved the paper and the manuscript has been suitably revised. I have no further specific comments.

7. PLOS authors have the option to publish the peer review history of their article (what does this mean?). If published, this will include your full peer review and any attached files.

Reviewer #1: No

Reviewer #3: No

Reviewer #4: **Yes: **Dr. Jedrzej Hoffmann, MD

---

## [Author Response · Author response to Decision Letter 1]

6 Oct 2021

Detailed answers to the editor comments:

1) In the abstract, limitation regarding a lack of long-term effect on inflammation and outcomes should be added, such as “long-term effect needs to be investigated” or “clinical studies are warranted to assess the long term effect”.

This sentence was added to the abstract, Conclusion section.

2) The conclusion in the abstract, “this short term effect may be associated with long term valvular durability” is not supported by your data and should be deleted.

This sentence was deleted.

3) The sentence, “an increase in WBC … was associated with higher rates of major …” (result section (clinical outcomes) and the sentence, “this response was also found to be associated with a worse 30 days outcome” (Discussion (first paragraph) should be de-emphasized since there is no difference or tendency in 30-day outcome (table 5).

The order of the sentence in the Discussion section, first paragraph, was inadequate and lead to confusion. Findings from table 4 and 5 wrote intercalates, provoke confusion. Now, we order in a more clear version. 

4) Small sample number should be included in the limitation.

We added a paragraph to describe the study limitations, including sample size.

Detailed answers to the reviewer #1 comments:

1. In table 2 and table 3, authors listed 24hr WBC, neutrophil, lymphocyte, and NLR. However, the values are a little different (ex: WBC 10.08±3.55 in table 2 vs 10.101±3.671). Do they calculate from the same groups???

The minuscular differences in the blood cell count related to the drop down of two patients excluded in table 3 because lack of 6 month follow-up lab values.

2. In discussion: “Our results are in agreement……..which probably can shed light this finding” may be deleted. 

This extended paragraph analyzed the core of our study integrated with the current published bibliography in the field. We see a legitimate discussion to include in the article. 

3. In discussion, “To the best of our knowledge…….heightened inflammatory response as compared to the BEV system” is good and reasonable to paper result. If author can list the table compare with nitinol concentration, D-dimer, fibrinogen…..so on between SEV and BEV, these biochemistry data will be more convincible and can more focus on inflammation response. That will be the real first report.

We have limited values of these propose parameters. Nitinol concentration was values were not measured at all.

4. How do authors select BEV or SEV methods for the patient treatment??? Authors didn’t answer directly. The selection bias may associate post-inflammation response. Please describe and clarity. 

This requires was answered previously.

5. Contrast and surgical time are very significant difference after 24hr postoperatively to preoperatively. Contrast amount and prolonged surgical time may induce cytokine secretion. Please compare with contrast and surgical time between SEV and BEV. It may explain the inflammation response different between groups.

In table 1 we demonstrated non-significant differences in the injected contrast material and procedural time between SEV and BEV. 

6. What about prior aortic valve replacement surgery between SEV and BEV (valve-in-surgical valve)??

We performed V-in-V in a few number of patients, 7 in total.

7. How many valves were used (rescued) during surgery if the first deployed valve was malfunctional (valve-in-valve)??

Rescue additional valve implantation migration related was performed in a small number of patients, 6 in total.

8. What about re-intervention rate between two groups (valve-in-valve)??

Again, the patient needed re-intervention after TAVI is very small.

---

## [Editor Report · Decision Letter 2]

11 Oct 2021

Differential systemic inflammatory responses after TAVI: The role of self versus balloon expandable devices

PONE-D-21-04417R2

Dear Dr. Blatt,

We’re pleased to inform you that your manuscript has been judged scientifically suitable for publication and will be formally accepted for publication once it meets all outstanding technical requirements.

Kind regards,

Michinari Nakamura, MD

Academic Editor

PLOS ONE
---

## [Editor Report · Acceptance letter]

14 Oct 2021

PONE-D-21-04417R2 

Differential systemic inflammatory responses after TAVI: The role of self versus balloon expandable devices 

Dear Dr. Blatt:

I'm pleased to inform you that your manuscript has been deemed suitable for publication in PLOS ONE. Congratulations! Your manuscript is now with our production department. 

Kind regards, 

on behalf of

Dr. Michinari Nakamura 

Academic Editor

PLOS ONE